# Two Korean Endemic *Clematis* Chloroplast Genomes: Inversion, Reposition, Expansion of the Inverted Repeat Region, Phylogenetic Analysis, and Nucleotide Substitution Rates

**DOI:** 10.3390/plants10020397

**Published:** 2021-02-19

**Authors:** Kyoung Su Choi, Young-Ho Ha, Hee-Young Gil, Kyung Choi, Dong-Kap Kim, Seung-Hwan Oh

**Affiliations:** 1Institute of Natural Science, Yeungnam University, Gyeongsan-si, Gyeongbuk-do 38541, Korea; choiks010@gmail.com; 2Department of Life Sciences, Yeungnam University, Gyeongsan-si, Gyeongbuk-do 38541, Korea; 3Forest Biodiversity Division, Korea National Arboretum, 415 Gwangneungsumogwon-ro, Soheul-eup, Pocheon-si, Gyeonggi-do 11186, Korea; yh0990@korea.kr (Y.-H.H.); botany7403@korea.kr (D.-K.K.); 4Department of Life Science, Gachon University, Seongnam-si, Gyeonggi-do 13120, Korea; 5DMZ Botanic Garden, Korea National Arboretum, 916-70, Punchbowl-ro, Haean-myeon, Yanggu, Gangwon-do 24564, Korea; warmishe@korea.kr; 6Research Planning and Coordination Division, Korea National Arboretum, 415 Gwangneungsumogwon-ro, Soheul-eup, Pocheon-si, Gyeonggi-do 11186, Korea; kchoi69@korea.kr

**Keywords:** chloroplast genome, *Clematis*, rearrangement, inversion, IR expansion, nucleotide substitution rate

## Abstract

Previous studies on the chloroplast genome in *Clematis* focused on the chloroplast structure within Anemoneae. The chloroplast genomes of *Cleamtis* were sequenced to provide information for studies on phylogeny and evolution. Two Korean endemic *Clematis* chloroplast genomes (*Clematis brachyura* and *C. trichotoma*) range from 159,170 to 159,532 bp, containing 134 identical genes. Comparing the coding and non-coding regions among 12 *Clematis* species revealed divergent sites, with carination occurring in the *petD*-*rpoA* region. Comparing other *Clematis* chloroplast genomes suggested that *Clematis* has two inversions (*trnH-rps16* and *rps4*), reposition (*trnL*-*ndhC*), and inverted repeat (IR) region expansion. For phylogenetic analysis, 71 protein-coding genes were aligned from 36 Ranunculaceae chloroplast genomes. Anemoneae (*Anemoclema*, *Pulsatilla*, *Anemone*, and *Clematis*) clades were monophyletic and well-supported by the bootstrap value (100%). Based on 70 chloroplast protein-coding genes, we compared nonsynonymous (*dN*) and synonymous (*dS*) substitution rates among *Clematis*, Anemoneae (excluding *Clematis*), and other Ranunculaceae species. The average synonymoussubstitution rates (*dS*)of large single copy (LSC), small single copy (SSC), and IR genes in Anemoneae and *Clematis* were significantly higher than those of other Ranunculaceae species, but not the nonsynonymous *substitution rates (dN)*. This study provides fundamental information on plastid genome evolution in the Ranunculaceae.

## 1. Introduction

The development of sequencing technology has enabled next-generation sequencing (NGS), which generates large amounts of data in a short time [1]. The increased availability of genomic data has led to the exploration of plant evolutionary history, including identifying hybrid origins [2], mapping organelles [3] or complete genomes [4], and tracing gene transfers among organelles [5]. The chloroplast (cp) is a self-replicating organelle in plants introduced through endosymbiosis and it plays a crucial role in photosynthesis [6]. The cp genome typically consists of a large single copy (LSC), a small single copy (SSC), and two inverted repeat (IR) regions. In angiosperms, cp genomes generally range from 120 to 160 kb in length and contain 110–130 genes. The gene content and gene order are highly conserved in most angiosperms [7,8], but different studies have reported changes in the cp genome, including gene loss [9,10], inversions or deletions [11,12], and the expansion or contraction of the IR regions [13,14]. Although the majority of IR regions in land plants range from 15 to 30 kb in length and contain four rRNA genes, five tRNA genes, and four coding genes, variation in locus- and lineage-specific structures has been reported [15,16]. The IR region has been considered to be involved in genome stability [17], and, therefore, suppressed synonymous and nonsynonymous mutations were observed in the IR region relative to the SC regions [18,19]. Although IR regions contribute to structural stability and are known to change conservatively, variable length variation has been observed. Large expansions (exceeding several kb) were reported in the IR regions of *Pelargonium* [20], *Berberis* [21], and *Asarum* [22]. In contrast, the contraction of the IR regions has been observed in legumes [23] and *Erodium* [24]. The variation in expansion and contraction in IR regions is correlated with the substitution rate of genes [5,14,15]. Although some studies have schematized their structure [25], no comprehensive research has been conducted, including only fragmented research in *Clematis* L. 

The family Ranunculaceae is within an early-diverging angiosperm lineage and comprises 59–62 genera and approximately 2500 species with a worldwide distribution [26,27]. The family is often utilized as a model system for research on plant evolution, given its high morphological diversity (e.g., fruit types and floral organization) [25,27]. The conventional cytologic classification recognizes three chromosome types: R (*Ranunculus*), T (*Thalictrum*), and C (*Coptis*). [28,29]. Other classifications have been proposed using morphological characters [26,30] and molecular data [27,31]. A recently published phylogenomic study using cp genome sequences reported that T-type chromosomal characteristics are important for the classification of Ranunculaceae [25]. Such genome studies evaluated genome diversity beyond estimating relationships. In the past decade, the cp genome analysis of Ranunculaceae revealed intracellular gene transfer (*infA, rpl32,* and *rps16*) in the tribe Delphinieae and genus *Thalictrum* [32]. In addition, the phylogenetic relationship and genomic structure of *Aconitum* spp. [33,34], *Hepatica* [35], and *Pulsatilla* [36] have been analyzed. However, there is still a need for in-depth research at the genus level. The *Clematis* genome has been directly published [37,38], as well as being used for other major studies [25,39,40]. Many studies have focused on *Aconitum* species, which are used in traditional herbal medicine in Asia [34,41,42]. Therefore, to the best of our knowledge, ours is the first study to perform genome-scale comparisons among *Clematis* species.

*Clematis*, commonly known as leather flowers, of Ranunculaceae comprise approximately 250–350 species, from lianas to subshrubs [43,44]. *Clematis* is a popular horticulture species in gardens, and various cultivars have been produced in Japan and China [45]. This genus occurs in almost all continents except Antarctica and shows great diversity in East Asia and North America [26,46]. Although several molecular studies of the *Clematis* tribe Anemoneae have been conducted and supported their monophyly, only some fragmentary sequences and partial intergenic spacers were used [47,48,49]. The *Clematis* cp genomes were used as related species for the study of other species. Therefore, there were no genomic comparative analyses among *Clematis* species [38,39]. Recently conducted phylogenetic analyses using complete cp genome sequences [50] provided important insight into two small genera, *Archiclematis* and *Naravelia*, that are closely related to *Clematis* and indicated that they should be included in *Clematis*. However, comparative genomic studies on the variation in structural diversity in *Aconitum* [5] and *Coptis* [25] in Ranunculaceae are still insufficient. 

In Korea, the genus *Clematis* comprises 22 species, including three endemic species (*C. brachyura*, *C. trichotoma,* and *C. fusca* var. *coreana*) [51]. An anatomy study conducted by Oh [52] separated each Korea endemic three species. However, there has never been a morphological and molecular phylogenetic analysis of the Korean *Clematis* species.

In the present study, we characterized the complete cp genomes of two Korean endemic *Clematis* species (*C. brachyura* and *C. trichotoma*). The aims of the present study were 1) to determine the molecular features of the genomes and 2) to compare *Clematis* and related taxa, focusing on structural variation such as inversions, rearrangements, and IR expansion–contraction. This study also calculated the substitution rates to trace the correlation between IR expansion and gene duplication in *Clematis*. Finally, protein-coding genes were used to reconstruct the phylogenetic relationships among *Clematis* and related species in Ranunculaceae. 

## 2. Results

### 2.1. General Chloroplast Genome Features

Two Korean endemic species, *C. brachyura* (MH104710) and *C. trichotoma* (MH104711), plastid genomes are 159,532 and 159,170 bp in length, respectively. The genomes of the two species had a typical quadripartite structure in which an LSC region (79,341 and 79,339 bp) and an SSC region (18,105 and 17,997 bp) were separated by two IRs (31,043 and 30,917 bp) (Figure 1 and Appendix A). The cp of *C. trichotoma* was the smallest among *Clematis* plastomes. In the two newly added cp genomes, 13 species of plastomes contained identical gene contents (134 genes, including 89 protein-coding, 8 rRNA, and 36 tRNA genes). *The infA* and *rpl32* genes were pseudogenized in all *Clematis* (Appendix A). 

The overall guanine–cytosine (GC) content of both *C. brachyura* and *C. trichotoma* was similar to 38% (LSC, 36.3%; SSC, 31.3%; IR, 42.1%) and 38% (LSC, 36.4%; SSC, 31.4%; IR, 42%), respectively. A total of 23 genes were duplicated in the IR regions, including 7 tRNA genes (*trnI-CAU, trnL-CAA, trnV-GAC, trnI-GAU, trnA-UGC, trnR-ACG,* and *trnN-GUU*), 4 rRNA genes (*5S rRNA, 4.5S rRNA, 23S rRNA,* and *16S rRNA*), and 12 protein-coding genes (*rps8, rpl14, rps16, rps3, rpl22, rps19, rpl2, rpl23, ycf2, ndhB, rps7,* and *rps12*). Sixteen genes (*atpF, ndhA, ndhB, petB, petD, rpl2, rpl16, rpoC1, rps12, rps16, trnA-UGC, trnG-UCC, trnI-GAU, trnK-UUU, trnL-UAA,* and *trnV-UAC*) contained one intron, and *clpP* and *ycf3* each contained two introns. Of the 16 intron genes, the intron sequence in *trnK-UUU* was the longest (2551 bp), and the intron sequence in *trnL-UAA* was the smallest (493 bp). 

### 2.2. Comparative Analyses 

The complete cp genome sequences of 12 *Clematis* species were plotted with mVISTA using the annotated *C. brachyura* cp genome as a reference (Figure 2). Based on the overall sequence identity indicated by the peaks and valleys among all 12 *Clematis* species, the results revealed that the LSC and SSC regions were divergent and the pairs of IR regions were highly conserved. Five non-coding regions (*trnT-psbD, atpB-rbcL, psbE-petL, pet D-rpoA,* and *ccsA-ndhD*) located in the SC regions were highly AT-rich and caused indels in *C. trichotoma* of more than 200 bp. High polymorphism was also observed in regions involved in pseudogenes (*infAψ* and *rpl32ψ*).

The IR and SC junctions of eight genes (including three kinds of pseudogenes) in 12 *Clematis* species were compared. The longest LSC was observed in *C. flabellata* (79,480), IR in *C. heracleifolia* (31,066), and SSC in *C. repens* (18,268) (Figure 3). Although protein-coding genes (*rpl36, ndhF, ycf1,* and *rps4*) showed less variation, non-functional pseudogenes revealed a high variation between species. In particular, the *C. repens* boundary was shown to be highly different from that of other plastomes.

The gene number and gene order were identical (Figure 3 and Appendix A) in all 12 *Clematis* chloroplast genomes and the shared characteristics of two non-functional pseudogenes in chloroplasts (Figure 4). The pseudogenization events of *infAψ* and *rpl32ψ* in the chloroplast genomes of *Clematis* were caused by different mechanisms, such as the loss of part of the coding region and frameshift mutation, respectively. Compared to the complete (functional) *infAψ* gene of *Helleborus*, a 180bp deletion was identified in *Clematis* species (Figure 4a). The shortest was observed in *C. repens* at 42 bp, including 14 amino acids. The tail regions of *infAψ* in four species (*C. brachyura*, *C. flabellata*, *C. loureiroana*, and *C. terniflora*), which share short genes, occurred due to point mutations in AT-rich regions and short insertions. The alignment of *rpl32ψ* genes was also influenced by the AT-rich regions of the 160 bp region. Compared with complete *Helleborus*, the *Clematis* species, which have short-type *rpl32ψ*, suffered point mutations in the 90 bp region (Figure 4b). 

The sliding window analysis conducted using DnaSP revealed highly variable regions in the 12 *Clematis* cp genomes (Figure 5). The average nucleotide diversity (Pi) over the entire cp genome was 0.00358, and four highly variable regions were identified based on a significantly higher Pi value of >0.015. The most variable region was the *petD/ropA* intergenic region with a Pi value of 0.02745. The alignment of the sequences of these regions detected distinctive variation between the four types (Appendix A). Polymorphism was detected in 12 species (*C. acerifolia, C. repens, C. macropetala, C. brachyura, C. terniflora, C. uncinata, C. flabellata, C. alternata, C. loureiroana,* and *C. tangutica*) due to substitution and short-region insertions/deletions. However, one group comprising three species (*C. brevicaudata, C. heracleifolia,* and *C. trichotoma*) showed a completely different sequence frame comprising approximately 100 bp. Other highly polymorphic sequences were located *in psbA*–*psbK* and *psbM*–*trnD* in the LSC regions. The highly variable regions of SSC were two intergenic regions, *ndhF/trnL-UAG* (Pi = 0.0208) and *ccsA/ndhD* (Pi = 0.01901), and one genic region: *ndhF* (Pi = 0.01514). 

### 2.3. Phylogenetic Relationships Analysis

For phylogenetic analysis, 71 protein-coding genes (51,496 bp) were aligned from 36 Ranunculaceae cp genomes, including 12 *Clematis* (Figure 6, Appendix A). The monophyly of the Anemoneae clade was highly supported (BS = 100) and the clade was closely related to the *Ranunculus* + *Halerpestes* clade. The *Clematis* clade is monophyletic, with high bootstrap support (BS = 100). The results of the present study confirmed that *Clematis* forms sister relationships with *Anemoclema*, *Pulsatilla*, and *Anemone*. The Korean endemic plants *C. brachyura* and *C. terniflora* formed one clade with high bootstrap support (BS = 100) and *C. trichotoma* is sister to *C. heracleifolia* and *C. brevicaudata* with high bootstrap support (BS = 100).

### 2.4. Intergeneric Genome Comparative Analyses

*Clematis* species have a well-conserved genomic structure and gene order (Appendix A). Here, *Clematis* species were compared with other Ranunculaceae genomes to identify genome rearrangements for each genus. In the tribe Anemoneae (*Pulsatilla, Anemone, Anemoclema,* and *Clematis*), inversion occurred in the LSC region (Figure 7). In *Anemoclema*, *Anemone*, and *Pulsatilla,* there were three inversions (Inversion I, *rps4;* Inversion II, *trnH-GUG/rps16;* and Inversion III, *trnS-GCU/trnS-GGA*). However, *Clematis* has different rearrangements and repositions. Comparisons between *Clematis* and other Anemoneae (*Anemoclema, Anemone,* and *Pulsatilla*) showed that *Clematis* has the same two inversions (Inversion I, *rps4*; Inversion II, *trnH-GUG/rps16*), whereas *Clematis* has repositioning (*trnL-UAA/ndhC*) and reinversion (*trnS-GGA/trnG-UCC*), and the gene order of the region (*trnS-GGA/trnG-UCC*) was similar to that of *Ranunculus* and *Thalictrum* (Figure 7).

The Anemoneae species showed a pattern of IR expansion, with the majority of expansion occurring in the LSC region. This region contains *rps4, rpl14, rpl16, rps3, rpl22, rps19*, and *rpl2*.

### 2.5. Comparison of Substituion Rates of Genes among Clematis, Anemoneae, and Ranunculaceae

The nucleotide substitution rates were compared among *Clematis*, Anemoneae (excluding *Clematis*), and Ranunculaceae (Appendix A). Seventy protein-coding genes were shared among 36 Ranunculaceae species. The average of synonymous substitution rates (*dS*) in the LSC, IR, and SSC genes of Anemoneae and *Clematis* were significantly higher than those of Ranunculaceae (*p* < 0.05), whereas the average of nonsynonymous substitution rates (*dN)* in the LSC, IR, and SSC genes of Anemoneae and *Clematis* were not significantly different (Figure 8).

The comparison of *dS* among cp genes showed that 34 genes and 11 genes in *Clematis* species were significantly higher than those in Ranunculaceae and Anemoneae, respectively (*p* < 0.05). Seventeen genes in Anemoneae had significantly higher *dS* values than in Ranunculaceae. Among the 70 plastid coding genes, 14 genes (*atpE, atpH, atpI, cemA, ndhJ, petA, petB, psbD, psbN, rpl36, rps7, rps11, rps18*, and *ycf3*) of the Anemoneae and *Clematis* species had significantly higher *dS* than other Ranunculaceae species (*p* < 0.05) (Appendix A). 

In *Clematis*, the *dN* for 25 genes was significantly higher than that of Ranunculaceae, and seven genes were significantly higher than those in Anemoneae species (*p* < 0.05). Thirteen genes (*matK, ndhC, ndhD, ndhF, ndhH, petA, psbK, rpl2, rpl36, rpoC1, rpoC2, rps11*, and *rps14*) of Anemoneae and *Clematis* had significantly higher *dN* than the other Ranunculaceae species (*p* < 0.05) (Appendix A).

To comprehensively examine the substitution rate of seven directly engaged genes in IR expansion and contraction, this study used seven genes distributed in IR-SC partial junctions (type A: Anemoneae; type B: Ranunculaceae without Anemoneae) (Figure 9). This analysis excluded the *rps19* gene because of its inconsistent location among the examined taxa. The *rpl2* gene, which was consistently located in the IR region, was used as a control condition, with the *dS* value being the lowest. The junction-localized gene *rpl22* had the highest *dS* value compared with the other genes (Figure 9). The *dS* value of *rps8* in type B was higher than that of *rps8* in type A (*p* < 0.05). Similarly, the *dS* values of *rpl14* and *rpl16* in type B were significantly higher than those in type A (*p* < 0.01). However, the *rps3* and *rpl2* genes did not show a distinct difference between types A and B. The *dN* of SC to the IR-shifted *rpl14* and *rpl16* genes in type A were higher than those of *rpl14* and *rpl16* in type B. However, the *rps3* and *rpl22* genes in type A were higher than those in Type B (Figure 9).

## 3. Discussion

*Clematis* contains well-recognized economical and horticultural species that are globally distributed [54]. In East Asia, there are 147 species (93 endemic) in China [55], 29 species (13 endemic) in Japan [56], and 22 species (4 endemic) on the Korean peninsula [52,57]. In the present study, we characterized the cp genomes of two Korean endemic *Clematis* species (*C. brachyura* and *C. trichotoma*) and compared them with those of related species in Ranunculaceae (Figure 1; Appendix A). Contrary to the theory of a conserved typical structures in the cp genome [5,22], structural variation was revealed in *Clematis*. Zhai et al. [25] found that *Clematis* had experienced four rearrangements compared with *Coptis*, which is an ancestral condition in Ranunculaceae and has a typical chloroplast structure. These inversions were indicated by short dispersed repeats or tRNAs, which play a role in promoting gene order changes by nonhomologous recombination [58]. Comparisons among *Clematis* species showed the largest genome size in *C. acerifolia* (159,552 bp) and the smallest in *C. macropetala* (159,647 bp). A pair of IRs exceeded 30,000 bp in all *Clematis* genomes, which was confirmed to be caused by six/seven genes introduced from the LSC. A pair of *infA* and *rpl32* genes was identified as pseudogenes (Appendix A and Figure 4). A pseudogenized event has been reported in Ranunculaceae and is inferred to involve genes from cp that lost their function after intercellular gene transfer to the nucleus [5]. After pseudogenization, the *infA* and *rpl32* genes suffered frameshift mutations and substitutions/deletions [33]. 

As for nonfunctional *infA* and *rpl32* in *Clematis,* notably we found two types that were different in length (Figure 4). To the best of our knowledge, this finding has not been discussed in previous studies, which might have been insufficient, given the availability of whole cp genomes [25]. The *infA* gene is known to be transcribed as polycistronic mRNA and is a constituent of the ribosomal protein (*rpl23*) operon [59], whereas *rpl32* plays a role in the ribosome structure. According to Millen et al. [60], the gene loss (e.g., pseudogenization) of *infA* independently occurred several times during evolution and might have been derived from the translocation of the gene to the nucleus. Similarly, the absence of the *rpl32* gene was identified in Thalictroideae (including *Aquilegia*, *Enemion*, *Isopyrum*, *Leptopyrum*, *Paraquilegia*, *Semiaquilegia*, and *Thalictrum*) [33]. Additionally, the transfer of *rpl32* to the nucleus has been reported in angiosperms [33]. However, there is no evidence for the transfer of *infA* and *rpl32* from the cp genome to the nuclear genome in *Clematis.* Further studies on the transcriptomes of these two genes should be conducted to clarify the effects of length variation in *Clematis*. 

The interspecies comparisons based on mVISTA (Figure 2) and nucleotide diversity (Figure 5) showed that the highly variable (low identity) region was mainly distributed in the SC regions compared with the IR region. Most highly variable regions were identified near the SC-IR junctions of intergenic spacers of *petD/rpoA* in LSC and *ndhF*/*trnL* in the SSC region. Notably, the highest nucleotide diversity (Pi = 0.02745) was found in the region *petD/rpoA* and showed highly distinct variation (Appendix A). Unlike the rest of the species, three species (*C. brevicaudata, C. heracleifolia,* and *C. trichotoma*) shared entirely different sequences comprising approximately 90 bp. *C. trichotoma,* a species endemic to South Korea, had the shortest length of inclusion compared with the other two species (*C. brevicaudata* and *C. heracleifolia*). This unique polymorphism might be the result of isolation from a common ancestor and a distinct differentiated lineage [61]. Although additional studies using comprehensive species are needed, this result suggests that the *petD/rpoA* region can be applied as a useful tool in phylogenetic inference or to study evolutionary history. The analysis of the endpoint in single-copy (SC) and IR regions is important because their length variation is caused by expansion and/or contraction [8]. The junction boundary results showed that functional groups (*rpl36, ndhF, ycf1,* and *rps4*) have similar tendencies, except for pseudogenized gene groups (*infAψ* and *ycf1ψ*). However, the junction endpoint (e.g., *petD/rpoA* and *ndhF/trnL*) was revealed to be over 70% AT-rich with various poly A sequences, and this poly-A region might play an important role in IR expansion or trigger variation [62]. In addition to AT-rich and dispersed repeats, substitution rates are also related to IR expansion [14,63,64,65]. 

Previous phylogenetic studies of *Clematis* were based on a few molecular markers. Xie et al. [32] showed that *Clematis* divided 10 clades based on the sampling of about 75 C*lematis* species using nr ITS and three chloroplast markers (*atpB-rbcL, psbA-trnH-trnQ,* and *rpoB-trnC* regions). However, this study did not include *C. trichotoma*. Lehtonen et al. [53] sampled 194 species which include *C. brachyura* and *C. trichotoma*. Lehtonene et al. [53] suggested that phylogeny divided genus *Clematis* into 12 clades ed, and *C. brachyra* and *C. trichotoma* were placed as clade C and clade K, respectively. This study showed *C. brachyura* was closely related to *C. terniflora*, which was also placed in clade C by a previous study [53]. The *C. trichotoma* was sister to *C. heracleifolia* and *C. brevicaudata*. Three species were placed in clade K by a previous study [53]. Thus, our results supported the relationship of genus *Clematis* by Lehtonene et al. [53].

Conformational changes in cp genomes (e.g., inversion, extension, contraction, and rearrangement) are major issues [66,67]. Previous studies have shown that *Clematis* and *Anemone* species have multiple inversions and transpositions that include many genes in the LSC region (Figure 7) [36,40]. However, these studies did not analyze the substitution rates. In the case of Geraniaceae, Weng et al. [68] suggested that plastid genome rearrangement was correlated with acceleration in *dN*. Most inversions in *Clematis* and Anemoneae occurred in the LSC region. Our data showed that the average *dS* in Anemoneae and *Clematis* LSC genes were significantly higher than that of *Ranunculus* LSC genes, whereas the *dN* was not significantly different. Weng et al. [68] suggested plastid genome rearrangements are correlated with the acceleration of *dN*. However, these were not the same as previously reported trends. 

Anemoneae, including *Clematis*, experienced IR expansion from six or seven protein-coding genes, which are usually located in the LSC region [25]. A similar case was also reported in *Tetracentron* and *Trochodendron* (Trochodendraceae) [69], of which five protein-coding genes (*rps8*, *rpl14*, *rpl16*, *rps3*, and *rpl22*) have shifted to IR regions. It was suggested that a pair of IR regions increased genetic stability (repair/maintenance efficiency) and thus played a role in the stability of genes positioned in IR as compared to SC regions. However, other studies have reported that IR does not contribute to genome stability in several genera (*Pelargonium,* [15]; *Erodium*, [24]; *Plantago*, [14]). Our results showed that the genes located in the IR regions had relatively low *dS* values; type A genes that experienced expansion of IR did not follow this trend (Figure 8). In other words, the expansion in *Clematis* did not support the hypothesis of the involvement of the IR regions in stability. It had reduced he DtNA repair/maintenance efficiency in the cp genome, which increased destabilization accompanied by recombination and substitution [15]. A distinct tendency was not confirmed in the *dS* values of the six genes in the extended region. Our results hypothesized that the *dS* of IR expansion is lower than that of the SC region, and the estimated value will reveal the increase/decrease direction. The *dS* (*rps8, rpl14,* and *rpl16*) values were consistent with expectations; however, two genes (*rps3* and *rpl22*) were found to be the same as type B or higher (Figure 8). These genes have experienced evolutionary history without direction, such as the locus-specific, IR-independent effects [15] shown in legumes [16], *Silene* [70], and *Pelargonium* [15]. 

The results of the present study enlarged the genomic data of endemic species on the Korean Peninsula. In particular, *Clematis* species have IR expansion and structural mutations among intergenic species. The IR structural variation provides valuable insight into the evolutionary history of cp genomes in Ranunculaceae. In addition, phylogenetic analysis and discovery of divergence hotspots revealed fundamental data for understanding the relationships among the genera and species of Ranunculaceae.

## 4. Materials and Methods

### 4.1. Taxon Sampling, DNA Extraction, Chloroplast Genome Sequencing, and Characterization

Fresh young leaves of *C. brachyura* and *C. trichotoma* were collected from wild individuals. Voucher specimens of two *Clematis* accessions were deposited in the Herbarium of the Korea National Arboretum (KH). Total genomic DNA was isolated using the DNeasy Plant Mini Kit (Qiagen Inc., Valencia, CA, USA). The quality of genomic DNA was measured using Nano Drop 2000 (Thermo Fisher Inc., Waltham, MA, USA), and quantity was checked using 1% agarose gel. Illumina paired-end libraries were constructed and sequenced on the MiSeq platform by Macrogen Inc. (Seoul, South Korea). A total of 8,572,072 and 7,505,088 reads of the 301-bp paired-end sequence (550 insert size) were generated from the sequencing libraries of *C. brachyura* and *C. trichotoma*, respectively. All the paired-end reads of each species were assembled de novo into draft contigs using Velvet v. 1.2.03 [71]. Thereafter, contigs were assembled into a circular complete genome with the reference genome of *C. terniflora* (NC_028000) using Geneious v. 10.2.2 [72]. Protein coding genes and rRNA genes were identified using the Dual Organellar GenMe Annotator [73] and tRNA genes were identified using tRNAscan-SE 2.0 [74]. The gene region and protein coding sequences were manually adjusted using Geneious v. 10.2.2 [72]. Two *Clematis* circular cp genome maps were drawn using OGDRAW [75].

### 4.2. Interspecific Genome Comparative Analyses

The complete cp genomes of *C. brachyura* and *C. trichotoma* from this study and cp genomes of 10 other *Clematis* species from GenBank were compared using mVISTA [76,77] with the LAGAN alignment program [78]. To compare the cp genomes of 12 *Clematis* species, the *C. brachyuran* cp genome was used as a reference. Major variations in the gene content or features of *Clematis* cp genomes were manually identified using Geneious v. 10.2.2 [72]. For accurate genome comparison, the gene annotation of 10 *Clematis* species was performed again with BLASTN, BLASTX, and tRNAscan-SE [74].

A DNA polymorphism analysis was performed using DNA Sequence Polymorphism (DnaSP) v6 [79] to calculate the nucleotide diversity (Pi) and to identify highly variable sites among *Clematis* cp genomes. Cp genome sequences were aligned using MAFFT implemented in Geneious v. 10.2.2 [72,80]. In the DNA polymorphism analysis, the window length was set to 800 bp and the step size was set to 200 bp.

### 4.3. Phylogenetic Analysis

In the phylogenetic reconstruction of the core Ranunculaceae group, 12 *Clematis* species, including two from this study, and a total of 17 genera and 36 taxa of Ranunculaceae were included (Appendix A). The cp genome of *Hydrastis canadensis* (KY085918) from GenBank was used as the outgroup because *Hydrastis* (Hydrastideae) is one of the most basal lineages of Ranunculaceae and has a sister relationship with the core Ranunculaceae and Coptideae [25]. The phylogenetic analysis data matrix was constructed using 76 commonly shared protein-coding genes from 36 Ranunculaceae cp genomes. Protein-coding genes were extracted and aligned using MAFFT [80]. Aligned protein-coding genes were concatenated using Geneious [72]. The program jModelTest 2 was employed to determine the optimal substitution model [81]. Maximum likelihood (ML) analyses were performed using RAxML v7.4.2, with 1000 bootstrap replicates using the selected best-fitting model: the GRT + I + G model [82]. 

### 4.4. Substitution Rate Estimation

Seventy protein-coding genes shared by 36 Ranunculaceae species (including 12 *Clematis* species) were extracted and aligned using MAFFT [80] (Appendix A). Phylogenetic analysis was performed using the ML method on RaxML [82]. To estimate the rates of nucleotide substitution, nonsynonymous and synonymous rates were calculated in PAML v.4.8 [83] using the CODEML option employing the F3 × 4 codon frequency model, and gapped regions were excluded with the “cleandata = 1” option. Rate estimations were performed on the following datasets: (1) concatenated genes for LSC, SSC, IR, and five genes (*rps8, rpl14, rpl16, rps3*, and *rpl22*) that are located in LSC or IR among Anemoneae (excluding *Clematis*), *Clematis*, and other Ranunculaceae species. (2) All the individual genes from plastid genes among Anemoneae (excluding *Clematis*), *Clematis*, and other Ranunculaceae species. (3) The five genes (*rps8, rpl14, rpl16, rps3*, and *rpl22*) that are located in LSC or IR between Anemoneae (including *Clematis*) and other Ranunculaceae species. Statistical analyses were conducted using R v. 3.4.2, and Bonferroni correction for comparison was applied.

## 5. Conclusions

In this study, the cp genomes of two Korean endemic species (*C. brachyura* and *C. trichotoma*) were assembled. As for angiosperms, the cp genome size, structure, and gene contents were highly conserved. However, this study demonstrates that *infA* and *rpl32* genes in *Clematis* were inferred to be pseudogenes. In addition, two inversions, reposition, and IR expansion were detected in the genus *Clematis*. The phylogenetic analyses of the genus *Clematis* are monophyletic, with 100% bootstrap values. *C. brachyura* and *C. terniflora* formed a clade with 100% bootstrap values and *C. trichotoma* was placed as a sister to *C. heracleifolia* and *C. brevicaudata* with 100% bootstrap values, which supports a previous phylogenetic study [66]. The comparative analysis of *Clematis* cp genes showed several variation hotspots. The IR expansions and rearrangements of the *Clematis* cp genome are not correlated with the acceleration in substitution rates. This study could be used for phylogenetic studies in *Clematis* and whole cp genome comparison.

## Figures and Tables

**Figure 1 plants-10-00397-f001:**
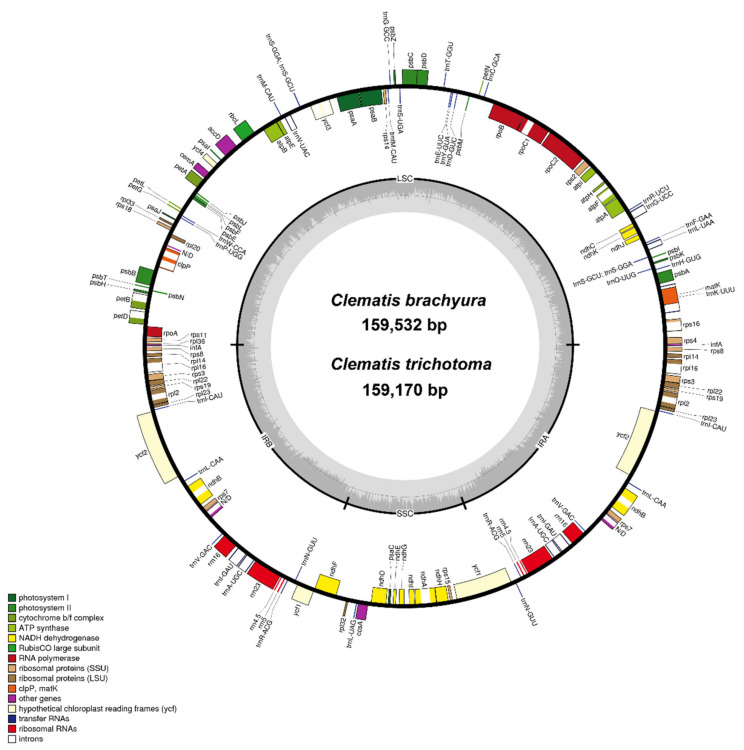
Chloroplast genome map for two *Clematis* species. Genes shown outside the circle are transcribed clockwise, whereas those inside the circle are transcribed counterclockwise. Genes belonging to different functional groups are colored. The dashed area in the inner circle indicates the GC content of the genome.

**Figure 2 plants-10-00397-f002:**
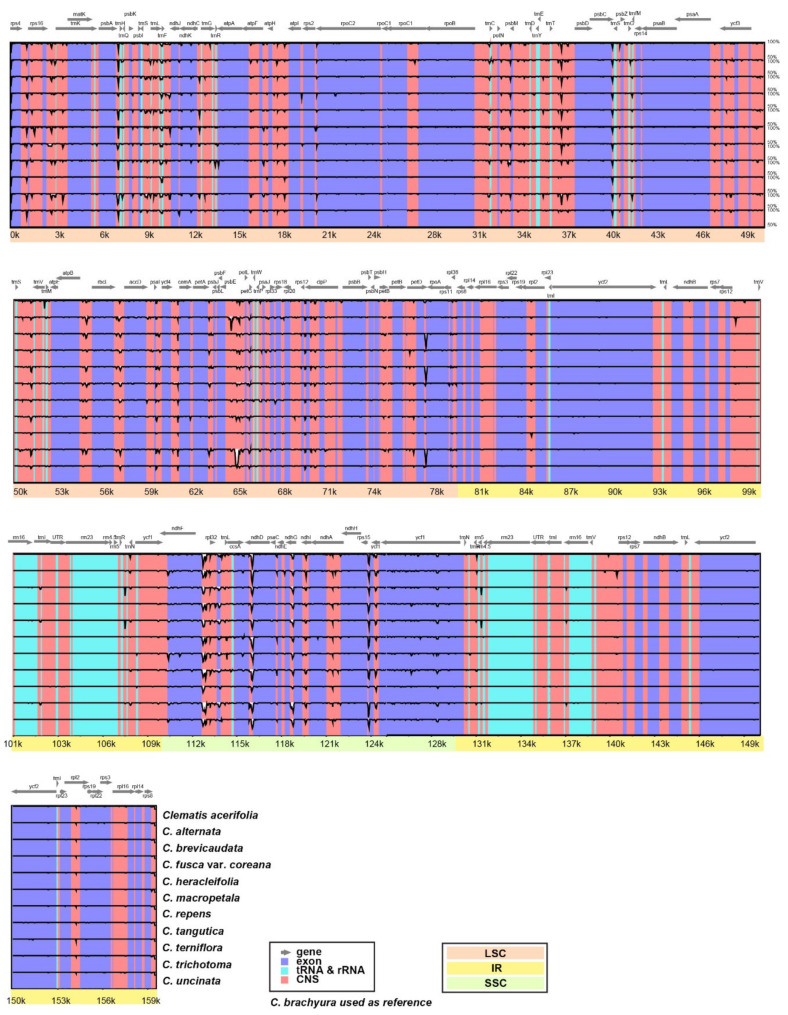
Chloroplast genome sequence alignment of 12 species of *Clematis* with *C. brachyura* used as a reference. The sequence identities were calculated and visualized in mVISTA. LSC, large single copy; SSC, small single copy; IR, inverted repeat.

**Figure 3 plants-10-00397-f003:**
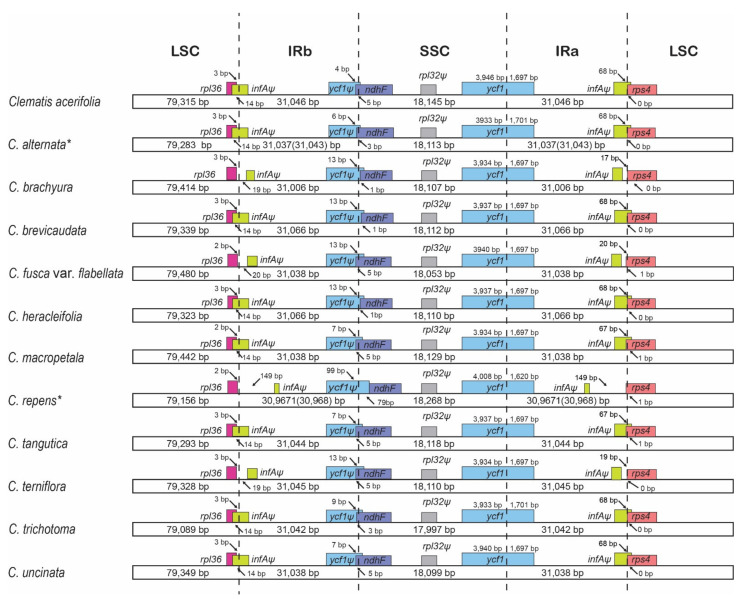
Comparisons of junctions (LSC to IR and IR to SSC IR region) among the chloroplast genomes of *Clematis*. * indicates different lengths between pairs of IRs. LSC, large single copy; SSC, small single copy; IR, inverted repeat.

**Figure 4 plants-10-00397-f004:**
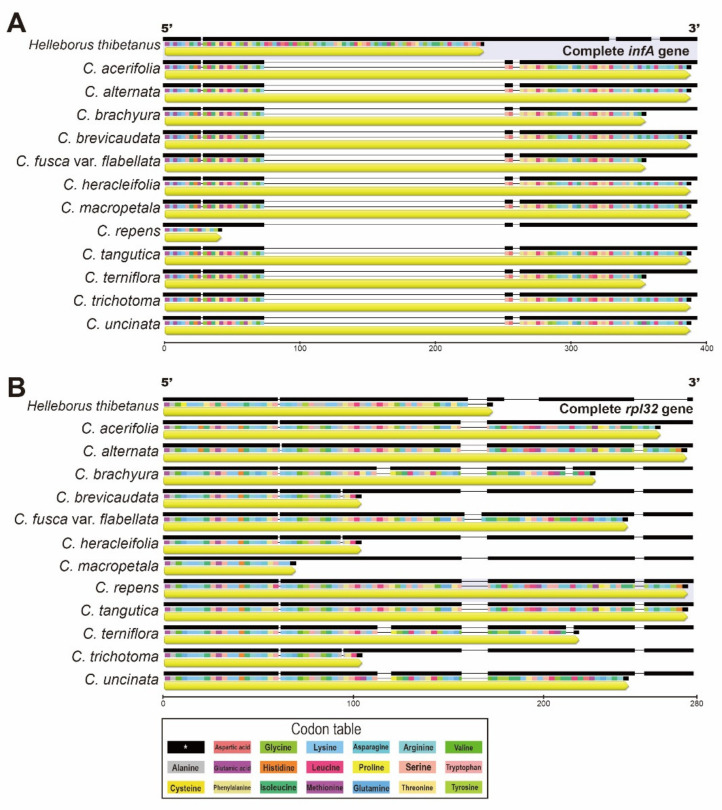
Alignment of two pseudogenes in *Clematis*. (**A**) *ψ infAψ*; (**B**) *ψ rpl32*.

**Figure 5 plants-10-00397-f005:**
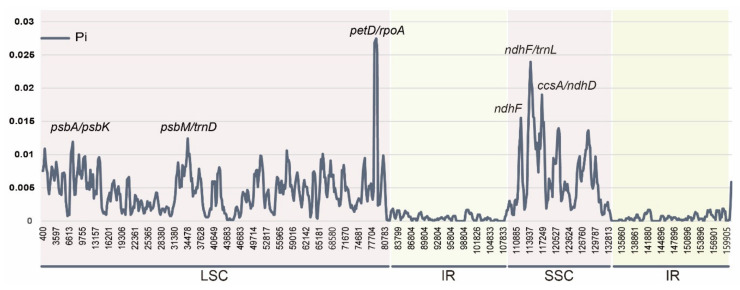
Sliding window analysis of the whole chloroplast genome for nucleotide diversity (Pi) compared among 12 *Clematis* species.

**Figure 6 plants-10-00397-f006:**
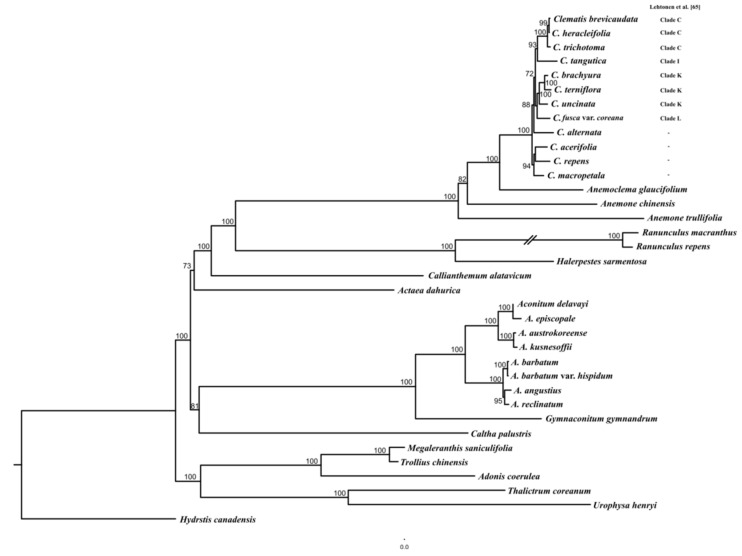
Maximum likelihood tree derived from 36 species and based on 77 concatenated protein-coding genes of Ranunculaceae. Bootstrap support values >70% are shown on the branches. Clade of *Clematis* followed Lehtonene et al. [53]. “-” did not include species by previous study [53].

**Figure 7 plants-10-00397-f007:**
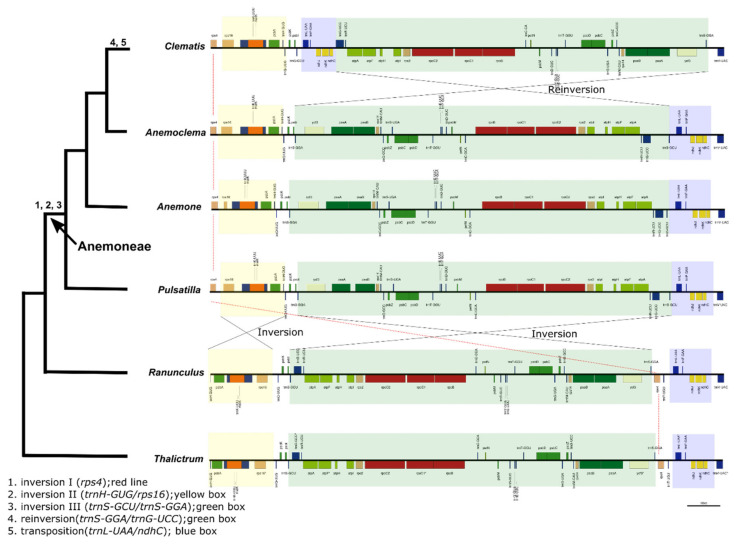
Patterns of rearrangement in the large single copy in *Ranunculus*. Rearrangement events are mapped on the branches.

**Figure 8 plants-10-00397-f008:**
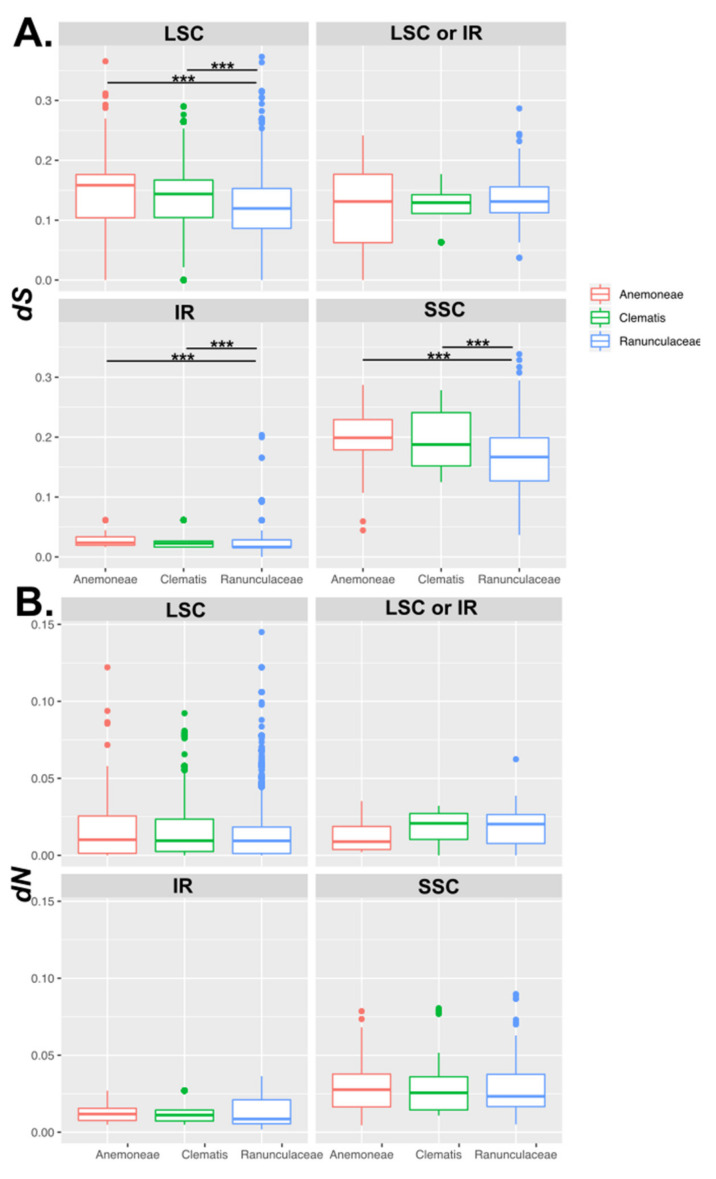
Comparison of synonymous (*dS)* and nonsynonymous (*dN*) substitution rates among Anemoneae, *Clematis*, and Ranunculaceae. (**A**) Comparison of *dS;* (**B**) Comparison of *dN*. LSC, large single copy region; IR, inverted repeat region; LSC or IR, genes were located in the LSC or IR regions; SSC; small single copy region. Asterisks indicate *p* < 0.05 (***).

**Figure 9 plants-10-00397-f009:**
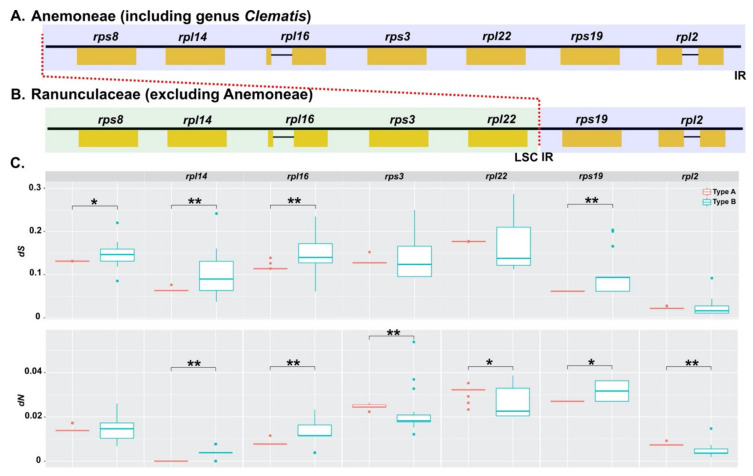
Extension of inverted repeat (IR) region in Anemoneae. IR expansion shown with red line. (**A**) Type A of IR expansion in Anemoneae; (**B**) type B of IR in Ranunculaceae excluding Anemoneae); (**C**) comparison of *dN* and *dS* between type A and type B genes. Asterisks indicate *p* < 0.05 (*) and *p* < 0.01 (**).

## Data Availability

The data presented in this study are openly available in GenBank.

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
