# Peer review of "Two Korean Endemic Clematis Chloroplast Genomes: Inversion, Reposition, Expansion of the Inverted Repeat Region, Phylogenetic Analysis, and Nucleotide Substitution Rates"

_plants, 2021, doi:10.3390/plants10020397_

Round 1

Reviewer 1 Report

The manuscript entitled “Two Korean Endemic Clematis Chloroplast Genomes: Inversion, Reposition, Expansion of the Inverted Repeat Region, Phylogenetic Analysis, and Nucleotide Substitution Rates” presents a study to determine the molecular features of the genomes, and to compare Clematis and related taxa, focusing on structural variation.

The work is properly organized and structured, with a clear understanding of the results, easy to follow and logically explained, with proper conclusions (avoiding any speculation) based on the data obtained. To my understanding, the methodology applied is adequate and complete.

The results presented are of scientific relevance, with interesting conclusions. Therefore, I recommend its publication in Plants.

Author Response

Thanks so much for your comments.

Reviewer 2 Report

Report for plants

Manuscript ID plants-1102036

The title: ‘Two Korean endemic Clematis chloroplast genomes: inversion, reposition, expansion of the inverted repeat region, phylogenetic analysis, and nucleotide substitution rates

General comments: -

I found the paper to be overall well prepared. The manuscript lacks some important parts. In general, it is well written, but I have some major comments.

Detailed comments: -

Keywords:

Please add phylogenetic analysis                                                    

Abstract

This section is missing the direct aim of this study please state the aim of the study in a clear and direct way.

Line 24: Please avoid using the personal pronouns (I, We,) and apply this rule throughout the manuscript.

Introduction

This section needs to be enriched and expanded by adding more background about the topic, especially about the Korean endemic Clematis and the phylogenetic analysis.

Results

Line 195-207

Phylogenetic relationship analysis part: (Please reword this part with enough citations and a clear explanation).

Discussion

Line 328-338: - Please Rewrite and clarify

Conclusion:

This section is missing: although it is not required in some article, this manuscript needs to have a solid conclusion to summarize and highlight the important findings

Author Response

Thanks for your comments and suggestions.

Reviewer 3 Report

Manuscript related to Clematis Chloroplast Genomes can be published in  Plants.
I have a few concerns:
Why the parameter in DNA polymorphism analysis, the window length was set to 800 bp, and the step size was set to 200 bp for this genome size?
Critically check for grammatical mistakes and English language mistakes.
Table 6 can be supplementary.
Figure 6 is blurred and difficult to predict.
Add the conclusion section.
I wonder if these results can be validated with a wet lab analysis?

Author Response

(The authors gave the same response as above.)

Round 2

Reviewer 2 Report

The Authors kindly have responded to all my comments and improved the manuscript. I see that the manuscript is now suitable for publication.